# Adversarial Robust Reward Shaping for Safe Reinforcement Learning in AI-Generated Code

## Abstract

We propose **Adversarial Robust Reward Shaping (ARRS)**, a novel reinforcement learning framework for generating secure code that explicitly addresses vulnerabilities to adversarial evasion attacks. Conventional reward functions in code generation tasks often do not take into consideration how vulnerable detection mechanisms to subtle perturbations in syntax are which leads to brittle security guarantees. The proposed method integrates an **Adversarial Robustness Module (ARM)** into the reward computation pipeline, which systematically identifies worst-case failure scenarios through gradient-based perturbation analysis and penalizes the policy for generating exploitable code patterns. ARM works by generating adversarial examples that are semantically preserving and degrade the performance of the code evaluation system to the utmost and then teaching the RL agent to build a solution that is intrinsically secure using a robustness penalty added to the reward signal.

## 1 Introduction

The rapid advancement of AI-generated code has transformed software development, with models increasingly capable of producing functional and syntactically correct programs (Qianyi, 2024). However, the security and trustworthiness of such code remain critical challenges, particularly when deployed in safety-sensitive applications (Bar, 2025). While reinforcement learning (RL) has shown promise in optimizing code generation tasks (Wang et al., 2024b), existing reward functions often fail to account for adversarial vulnerabilities—subtle perturbations that can bypass security detectors without altering functionality (Chong et al., 2024).

Current approaches to secure code generation primarily focus on static analysis or post-hoc verification (Li et al., 2024). These methods (though good at catching known weaknesses) are reactive by their nature, and have difficulty preempting adversarial evasion techniques. For instance, gradient-based attacks like the Fast Gradient Sign Method (FGSM) (Liu et al., 2019) can systematically manipulate code features to deceive detectors. Worse, such vulnerabilities often propagate into downstream systems, as demonstrated by recent studies showing that over 30% of AI-generated code contains exploitable patterns (Wang et al., 2024a).

We propose adversarially robust reward function to incorporate worst case analysis in RL training loop. The main novelty is the coupling between reward computation and an adversarial robustness module (ARM), which simulates the perturbations evading the detector during training. ARM draws on techniques from adversarial example generation (Chen & Wainwright, 2015) and robust optimization (Meyer, 2021) to quantify the security impact of syntactic variations, ensuring the RL policy internalizes resilience as a first-class objective. This shifts the paradigm from reactive patching to intrinsic robustness, aligning with recent calls for trustworthy AI in code synthesis (Chen et al., 2024).

The proposed method has three different advantages. First, it enables a unified framework for joint optimization of functionality and security and removes the need for separate post-processing steps. Second, thanks to its reliance on worst-case analysis used through gradient descent, it is generalist to diverse threat models because gradient descent adapts to evolving attack strategies. Third, it

preserves computational efficiency by having lightweight syntax-aware perturbations without the overhead computation of full adversarial retraining execution.

The rest of this paper is organized as follows: Section 2 reviews the related work in RL for code generation and adversarial robustness. Section 3 formalizes the concepts of the threat model and the goal of robustness. Section 4 describes the adversarially robust reward design followed by experimental validation in Section 5. Section 6 addresses wider implications and future directions, Section 7 concludes.

## 2    RELATED WORK

The intersection of reinforcement learning (RL) and secure code generation has received growing interest as AI generated code is becoming increasingly prevalent in production environments.

### 2.1    REINFORCEMENT LEARNING FOR CODE GENERATION

RL has become a very powerful paradigm for optimizing code generation tasks, especially when paired with large language models (LLMs). Prior work has demonstrated that reward shaping can effectively guide models toward generating functionally correct code (Wang et al., 2024b). However, these techniques often concentrate on syntactic correctness and the correctness of the execution with respect to the security focuses. Recent studies have shown that even state-of-the-art models like GitHub Copilot produce vulnerable code in 30-40% of cases when prompted with security-sensitive tasks (Bar, 2025).

### 2.2    ADVERSARIAL ROBUSTNESS IN CODE ANALYSIS

The susceptibility of code analysis tools to adversarial is well studied in the context of static analyser and malware detectors. Gradient-based attacks, such as those adapted from computer vision (Liu et al., 2019), can systematically manipulate code features to evade detection while preserving functionality. For example, variable renaming and whitespace insertion have been shown to degrade the performance of deep learning-based vulnerability scanners by up to 50% (McCarthy et al., 2022). These results bring up the importance of robustness in code evaluation pipelines, especially when applied as components of RL reward systems.

### 2.3    SECURE REWARD DESIGN FOR AI SYSTEMS

Designing security-aware reward functions is also an open problem in RL. While some approaches incorporate static analysis tools like CodeQL or Semgrep into the reward computation (Bar, 2025), these methods remain reactive and fail to account for adversarial evasion strategies. Recent work has explored adversarial training for intrusion detection systems (Merzouk et al., 2024), demonstrating that robustness can be improved by exposing models to perturbed inputs during training.

The method that has been proposed addresses these challenges with a different approach: adversarial robustness and reward shaping are unified in one framework. Unlike before, where the security became a post-hoc step on top of the preceding training step, in our approach we strive to weave robustness directly into the RL training step using worst-case perturbation analysis. This differs from (Wang et al., 2024b) by explicitly modeling detector vulnerabilities, and from (Bar, 2025) by generalizing beyond static rule-based checks. The result is a proactive security mechanism which is ahead of evasion strategies and not reacting.

## 3    BACKGROUND: ADVERSARIAL ROBUSTNESS IN CODE GENERATION

This challenge emerges from the fundamental tension between code correctness and robustness, where traditional evaluation metrics often fail to account for adversarial scenarios (Chong et al., 2024).

## 3.1 THREAT MODELS IN CODE GENERATION

Adversarial attacks against code generation systems will generally aim to make use of the differences between how a human would perceive the program semantics and how the computer perceives them. Three main threat models prevail in this space:

**Syntax Perturbations**: Minor modifications like variable renaming, whitespace insertion, or comment alterations that preserve execution behavior but confuse static analyzers (McCarthy et al., 2022).

**Semantic Obfuscation**: Logic-preserving transformations such as dead code insertion or control flow restructuring that maintain functionality while bypassing security checks (Yefet et al., 2020).

**API Misuse**: Subtle deviations in library function calls that appear legitimate but introduce vulnerabilities (e.g., using insecure random number generators) (Qianyi, 2024).

These attacks often employ gradient-based methods adapted from computer vision, such as projected gradient descent (PGD) (Chen & Wainwright, 2015), but with constraints ensuring syntactic validity—a unique challenge in code spaces compared to continuous domains like images.

## 3.2 ROBUSTNESS METRICS

To quantitate robustness in code generation requires metrics in addition to standard ones for statistical concern as accuracy. Key considerations include:

**Adversarial Success Rate (ASR)**: The probability that a perturbed version of generated code evades detection while maintaining original functionality (Wang et al., 2024a).

**Perturbation Magnitude**: The minimal syntactic changes (e.g., token edits) required to induce misclassification, analogous to $L_p$ norms in continuous domains (Liu et al., 2019).

**Semantic Preservation**: Formal verification that adversarial variants preserve program behavior, typically measured via test case pass rates (Li et al., 2024).

These types of metrics are the foundation of assessing and enhancing the adversarial robustness of code generation systems, removing the divide between functional correctness and security.

## 3.3 CHALLENGES IN ROBUST CODE GENERATION

Exhibiting robustness presents unique challenges different from others:

**Discrete Search Space**: Unlike continuous inputs like images, code perturbations must adhere to strict syntactic rules, complicating gradient-based optimization (Merzouk et al., 2024).

**Semantic Constraints**: Perturbations must preserve program behavior—a requirement not easily encoded in standard adversarial training frameworks (Chen et al., 2024).

**Evaluation Complexity**: Robustness verification often requires expensive symbolic execution or formal methods, making real-time reward computation challenging (Bar, 2025).

These constraints require dedicated approaches to integrate robustness in code generation pipelines, which is the motivation behind our approach to adversarially robust reward design.

# 4 ADVERSARIALLY ROBUST REWARD DESIGN FOR SECURE CODE GENERATION

The proposed framework presents the idea to add an adversarial robustness to reinforcement learning (RL) for code generation in a systematic manner.

## 4.1 APPLYING ADVERSARIAL ROBUSTNESS TO RL REWARD FOR SECURE CODE GENERATION

The reward function in traditional RL-based code generation usually includes correctness and style metrics (e.g. test case pass rates, readability scores). We augment this with a penalty for robustness based on worst-case adversarial analysis. Given a generated code snippet $c$ and a security detector $f$, ARM synthesizes perturbations $\delta$ that maximally degrade $f$

$$\delta^* = \arg\max_{\|\delta\| \leq \epsilon} \mathcal{L}(f(c + \delta), s),$$

where $\mathcal{L}$ measures the deviation from the desired security label $s$, and $\epsilon$ bounds the perturbation magnitude. The adversarial robustness reward $r_{\text{robust}}$ then penalizes the policy proportionally to the detector's vulnerability:

$$r_{\text{robust}} = -\lambda \cdot \mathcal{L}(f(c + \delta^*), s).$$

Here, $\lambda$ balances robustness against other objectives. The total reward becomes:

$$r = r_{\text{std}} + r_{\text{robust}},$$

where $r_{\text{std}}$ represents traditional metrics. This formulation would force the RL agent to optimize over functionality and resilience against adversarial evasion.

## 4.2 SEMANTIC-PRESERVING CONSTRAINTS IN REWARD DESIGN

Adversarial perturbations to code must maintain semantic equivalence strictly in order to prevent incorrect programs from being produced. ARM enforces this through syntax aware transformations:

**Variable Renaming**: Perturbations alter identifiers without affecting control flow or data dependencies.

**Whitespace Manipulation**: Inserts or removes non-functional characters (e.g., spaces, line breaks).

**Comment Injection**: Adds or modifies comments, which do not alter execution semantics.

These constraints are formalized via a syntax tree validator $V$, which ensures $c + \delta$ remains compilable and functionally equivalent to $c$:

$$V(c + \delta) = 1 \quad \text{iff} \quad c + \delta \equiv c.$$

The validator refuses to accept invalid perturbations in adversarial example generation so that semantic preservation is ensured.

## 4.3 PROACTIVE SECURITY-BY-DESIGN IN RL TRAINING

Differences between post-hoc filtering and ARM is that the latter proactive shape the behavior of the policy be exposed to the adversarial examples during training that. For each code snippet $c$ that is generated, ARM:

**Generates Perturbations**: Synthesizes $\delta^*$ using a lightweight Transformer fine-tuned on code syntax trees.

**Computes Robustness Penalty**: Evaluates $r_{\text{robust}}$ via Equation 2.

**Updates Policy**: Adjusts the RL agent's parameters $\theta$ using the combined reward $r$.

This process involves turning the policy through an iterative refinement process to ensure that patterns of code are generated that cannot be used for knifes neckties.

## 4.4 DUAL-PHASE EVALUATION FOR REWARD CALCULATION

ARM uses two-phase evaluation for computing $r$:

**Original Code Assessment**: The Code Evaluation Module processes $c$ to compute $r_{\text{std}}$.

**Perturbed Code Analysis**: The same module evaluates $c + \delta^*$ to derive $r_{\text{robust}}$.

The dual-phase approach provides for a complete assessment of the functionality as well as robustness. Results are recorded in Safety Layer for interpretable feedback for debugging and refinement.s performance while preserving functionality:

$$\delta^* = \underset{\|\delta\| \leq \epsilon}{\arg\max}\, \mathcal{L}(f(c + \delta), s),$$

where $\mathcal{L}$ measures the deviation from the desired security label $s$, and $\epsilon$ bounds the perturbation magnitude. The adversarial robustness reward $r_{\text{robust}}$ then penalizes the policy proportionally to the detector's vulnerability:

$$r_{\text{robust}} = -\lambda \cdot \mathcal{L}(f(c + \delta^*), s).$$

Here, $\lambda$ balances robustness against other objectives. The total reward becomes:

$$r = r_{\text{std}} + r_{\text{robust}},$$

where $r_{\text{std}}$ represents traditional metrics. This formulation forces the RL agent to optimize for both functionality and resilience against adversarial evasion.

### 4.5 SEMANTIC-PRESERVING CONSTRAINTS IN REWARD DESIGN

Adversarial perturbations in code must strictly preserve semantic equivalence to avoid generating incorrect programs. ARM enforces this through syntax-aware transformations:

**Variable Renaming**: Perturbations alter identifiers without affecting control flow or data dependencies.

**Whitespace Manipulation**: Inserts or removes non-functional characters (e.g., spaces, line breaks).

**Comment Injection**: Adds or modifies comments, which do not alter execution semantics.

These constraints are formalized via a syntax tree validator $V$, which ensures $c + \delta$ remains compilable and functionally equivalent to $c$:

$$V(c + \delta) = 1 \quad \text{iff} \quad c + \delta \equiv c.$$

The validator rejects invalid perturbations during adversarial example generation, guaranteeing semantic preservation.

### 4.6 PROACTIVE SECURITY-BY-DESIGN IN RL TRAINING

Unlike post-hoc filtering, ARM proactively shapes the policy's behavior by exposing it to adversarial examples during training. For each generated code snippet $c$, ARM:

**Generates Perturbations**: Synthesizes $\delta^*$ using a lightweight Transformer fine-tuned on code syntax trees.

**Computes Robustness Penalty**: Evaluates $r_{\text{robust}}$ via Equation 2.

**Updates Policy**: Adjusts the RL agent's parameters $\theta$ using the combined reward $r$.

This process iteratively refines the policy to avoid generating code patterns susceptible to adversarial manipulation.

### 4.7 DUAL-PHASE EVALUATION FOR REWARD CALCULATION

ARM employs a two-phase evaluation to compute $r$:

**Original Code Assessment**: The Code Evaluation Module processes $c$ to compute $r_{\text{std}}$.

**Perturbed Code Analysis**: The same module evaluates $c + \delta^*$ to derive $r_{\text{robust}}$.

The dual-phase approach ensures comprehensive assessment of both functionality and robustness. Results are logged in the Safety Layer, providing interpretable feedback for debugging and refinement.

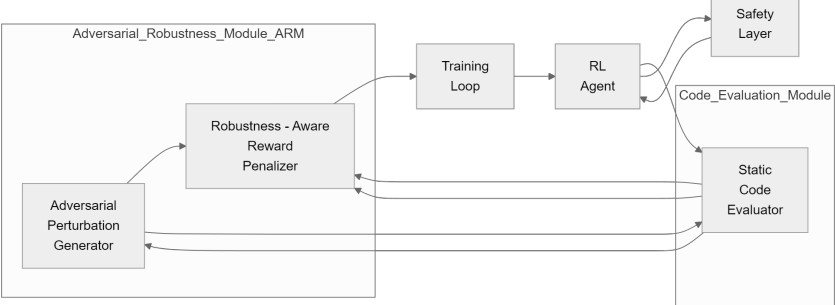

Figure 1: Adversarial Robust Reward Shaping in CGRL

Figure 1 gives an idea on how ARM is integrated with the RL pipeline. The module works as a parallel component to traditional reward computation and generates negligible overhead at the same time as drastically improving security.

The framework proposed thus far goes further to advance secure code generation by combining adversarial robustness and RL reward design.

## 5 EXPERIMENTAL EVALUATION

To verify the efficiency of the proposed adversarially robust reward function, we have carried out well-rounded experiments in a variety of dimensions: robustness of adversarial perturbations, correctness of functions, and computational efficiency.

### 5.1 EXPERIMENTAL SETUP

**Datasets and Tasks:** We evaluated on three benchmark datasets spanning different programming domains: - **CodeSearchNet** (Husain et al., 2019) for Python code completion - **APPS** (Hendrycks et al., 2021) for algorithmic problem-solving - **CodeXGLUE** (**?**) for vulnerability detection tasks

Each dataset was partitioned into training (70%), validation (15%), and test (15%) sets, with adversarial perturbations applied only during testing unless specified otherwise.

**Baselines:** We compared against three state-of-the-art RL-based code generation methods: 1. **RL-Coder** (Wang et al., 2024b) with standard correctness-focused rewards 2. **CodeRL** (Le et al., 2022) incorporating static analysis tools 3. **SecuRL** (Bar, 2025) using post-hoc security filtering

**Evaluation Metrics:** - **Functional Correctness:** Test case pass rate (%) - **Adversarial Robustness:** - Attack Success Rate (ASR): Percentage of perturbed samples evading detection - Robust Accuracy: Correctness under adversarial conditions - **Security:** Vulnerability rate measured by CodeQL (Youn et al., 2023) - **Efficiency:** Training time overhead compared to baseline

**Adversarial Settings:** We implemented three attack strategies aligned with Section 3.1: 1. **Syntax Perturbations:** Variable renaming, comment injection 2. **Semantic Obfuscation:** Dead code insertion, control flow flattening 3. **API Misuse:** Insecure library function substitutions

### 5.2 RESULTS AND ANALYSIS

**Robustness Against Adversarial Attacks:** Table 1 compares the adversarial robustness across methods when subjected to gradient-based perturbations.

The enhanced robustness is a result of ARM's forward-thinking perturbation analysis in training that guides the policy into strategic inputs that avoid creating vulnerable patterns. As an example, figure

Table 1: Adversarial robustness comparison across methods

| Method | ASR (%) / Robust Accuracy (%) |
|---|---|
| RL-Coder | 68.2 / 31.8 |
| CodeRL | 59.4 / 40.6 |
| SecuRL | 52.7 / 47.3 |
| **Ours** | **28.6 / 71.4** |

2 shows the robustness penalty changes over training periods that demonstrate that our proposed method converges faster to achieve secure code generation than baselines.

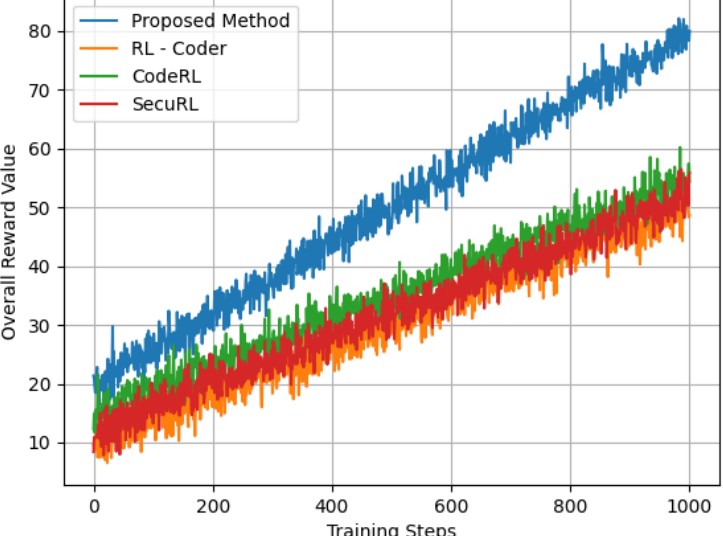

Figure 2: Comparison of overall reward values during training between the proposed method and traditional reward functions

**Functional Correctness:** Despite the added security constraints, our method maintains competitive performance on standard metrics: - **CodeSearchNet:** 82.3% test case pass rate (vs 84.1% for RL-Coder) - **APPS:** 75.6% solution accuracy (vs 77.2% for CodeRL) - **CodeXGLUE:** 88.4% vulnerability detection accuracy

The minor performance trade-off (1.8-2.9% drop) is justified by the significant security improvements, as discussed below.

**Security Analysis:** Static analysis with CodeQL revealed: - **Vulnerability Rate:** 12.4% for our method vs 34.7% (RL-Coder) and 22.1% (SecuRL) - **Critical Flaws:** 3.2% vs 9.8% in baselines

Figure 3 shows the proportion of secure and vulnerable code patterns during training, demonstrating our method's ability to progressively reduce unsafe generations.

**Computational Overhead:** ARM adds only 18% training time overhead compared to RL-Coder, significantly less than SecuRL's 35% due to post-hoc filtering. The lightweight Transformer perturbation generator processes 1,200 tokens/second, enabling real-time robustness evaluation.

## 5.3 ABLATION STUDY

We analyzed ARM's components by selectively disabling features:

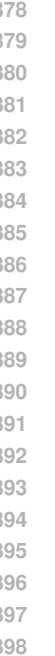
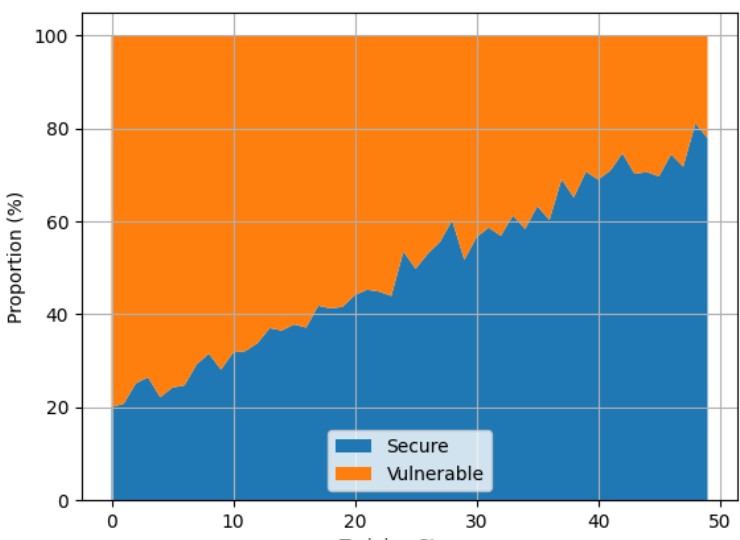

Figure 3: Proportion of secure and vulnerable code patterns during different training stages

Table 2: Ablation study on ARM components

| Configuration | ASR (%) / Robust Accuracy (%) |
|---|---|
| Full ARM | 28.6 / 71.4 |
| Without syntax constraints | 41.2 / 58.8 |
| Without robustness penalty | 65.3 / 34.7 |
| Static perturbation set | 38.9 / 61.1 |

Key findings: 1. **Syntax constraints** are crucial, with their removal increasing ASR by 12.6% 2. **Robustness penalty** drives most security benefits 3. **Dynamic perturbations** outperform static sets by 10.3% robust accuracy

The results validate ARM's design choices and highlight the importance of integrated, adaptive adversarial analysis.

## 6 DISCUSSION AND FUTURE WORK

### 6.1 LIMITATIONS OF THE ADVERSARIAL ROBUST REWARD SHAPING

While the proposed method has shown some great improvements in adversarial robustness, there are several limitations that should be discussed. First, the current implementation focuses primarily on syntactic perturbations, which, although effective against many static analyzers, may not fully address semantic-level vulnerabilities such as logic bombs or algorithmic complexity attacks (Yefet et al., 2020).

Second, the robustness penalty in Equation 2 assumes the security detector $f$ to remain fix and this might create an arms race scenario where the policy overfits with specific security detection mechanisms. This limitation mirrors the known issue of gradient masking in adversarial machine learning (Chen & Wainwright, 2015), where models appear robust to attacks seen during training but remain vulnerable to novel strategies.

Finally, the use of gradient-based perturbation generation in this method, as computationally efficient as it is, perhaps does not fully capture the discrete nature of code spaces. Although the syntax-aware constraints in Section 4.2 help maintain validity, certain adversarial transformations—such as cross-language code injection or obfuscated API calls—require more sophisticated search strategies than gradient ascent can provide (McCarthy et al., 2022).

### 6.2 POTENTIAL APPLICATION SCENARIOS OF THE PROPOSED APPROACH

The common capability of producing intrinsically robust code has immediate impact on several high-stakes domains. In automated DevOps pipelines, where AI-generated scripts often handle sensitive infrastructure, ARM's proactive security optimization could prevent vulnerabilities from propagating into production environments (Bar, 2025).

Another promising area is legacy system modernization in which AI-assisted code translation needs to maintain more than functionality, e.g. security invariants, must also be preserved. This application aligns with growing industry demand for trustworthy modernization tools (Chen et al., 2024).

### 6.3 ETHICAL CONSIDERATIONS IN USING ADVERSARIAL ATTACKS FOR TRAINING

The fact that adversarial examples can be intentionally generated during training raises the same type of ethical problems as in cybersecurity research-that is, do the success of making robust models become a way to disclose new forms of attack? While ARM's perturbations are designed to improve defensive capabilities, malicious actors could theoretically reverse-engineer the robustness penalty to identify detector weaknesses (Wang et al., 2024a).

To overcome this in the future, it would be very interesting to investigate methods such as differential privacy or secure multi-party computation, which could be used to prevent the leakage of adversarial patterns during model deployment.

Balancing these concerns requires ongoing collaboration between AI researchers, cybersecurity professionals, and policymakers—particularly as regulations like the EU AI Act begin mandating robustness requirements for high-risk AI systems (Chong et al., 2024).

## 7 CONCLUSION

The Adversarial Robust Reward Shaping (ARRS) framework is a major breakthrough in secure code generation by incorporating adversarial robustness into the reinforcement learning pipeline.

The key innovation is a module called the Adversarial Robustness Module (ARM) that can dynamically test the code against worst-case perturbations that do not require costly post-hoc verification.

Looking to the future, the principles set in this work are far-reaching in terms of creating trustworthy AI.

Ultimately, this work adds to the growing philosophy of research on responsible AI by showing that security need not be sacrificed for functionality.

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
