# OpenReview forum: "Adversarial Robust Reward Shaping for Safe Reinforcement Learning in AI-Generated Code"
_ICLR.cc/2026/Conference — Submitted to ICLR 2026_

### Official Review · Reviewer_82Rr · 2025-10-25

**Soundness:** 2
**Presentation:** 1
**Contribution:** 1
**Rating:** 2
**Confidence:** 3

**Summary:**

This paper proposes Adversarial Robust Reward Shaping, which focuses on LLM generated code's vulnerability towards subtle syntax perturbations. The propose algorithm introduces perturbations to sample adversarial cases and train an model. The paper conducts a rudimentary experiment on the topic and shows a promising start.

**Strengths:**

The idea of using adversarial learning for code LLMs and security vulnerability is a field that still have questions yet to be explored. The paper is a promising start in this direction.

**Weaknesses:**

While the paper is a sound start, I think the idea has to be developed further and more thoroughly experimented and validated to be accepted.

First, using an gradient based adversarial learning against gradient based attacks is an interesting and sound approach. However, however, gradient based method is not the only way in which LLM's vulnerability has been approached. For example, PAIR (Prompt Automative Iterative Refinement, Chao et al, 2023) uses an LLM to attack another LLM. While is is not expected that a single paper should all kinds of LLM attacks, it would be better to more clearly define the scope and the boundaries of the research.

In a related note, I think the paper could cite more related works in the field. While the paper cites some notable papers in the Code LLM, there's a wider breadth of research going on in the neighboring jailbreaking field. For example, while the paper cited Computer Vision research for gradient based approaches, Gradient Coordinate Gradient, is a jailbreaking technique used to exploit the security weakeness of LLMs and therefore would provide a more relavant example to a LLM research. Examples like this would provide a more concrete foundation for the research topic of the paper.

I also found the paper rather difficult to read and several details seemed to be missing. For example, it is not clear how ARM is updated during training to remain challenging and adversarial. Going as early as Robust Adversarial RL (Pinto et al, 2017), how to update the adversarial example generator has been an important part of adversarial robustness by worst-case scenarios. Also, it is difficult to reproduce the results of the paper from the main submission alone, as several details, including training hyperparameters and LLM model in question, seems to be missing.

Finally, there stylistic improvements that can be made. For example, in line 193, most conference submissions prefer equations to be center-aligned with equation number.

**Questions:**

For major changes to the scoring decisions, I would like need to see the following

- More comprehensive comparison with related works
- A better description & explanation on how the algorithm works
- Clearing up experiment details

---

### Official Review · Reviewer_7uKR · 2025-10-30

**Soundness:** 1
**Presentation:** 2
**Contribution:** 2
**Rating:** 2
**Confidence:** 4

**Summary:**

The paper introduces **Adversarial Robust Reward Shaping (ARRS)** for code-generation RL. A module ARM generates *semantics-preserving* perturbations $\delta^*$  to a produced snippet (c) that maximally degrade a security detector (f), and adds a robustness penalty to the reward. Perturbations are claimed to preserve semantics via syntax-aware edits (renaming, whitespace, comments) validated by an AST-based checker, and experiments on CodeSearchNet/APPS/CodeXGLUE report lower ASR and ~18% overhead.

**Strengths:**

The idea of *folding worst-case robustness directly into the reward* is conceptually appealing and well-motivated for code, where discrete, semantics-constrained perturbations matter.

**Weaknesses:**

* My first concerin is there is **no usable definition or concrete examples of “adversarial examples”**. The paper lists categories (renaming, whitespace, comments) but does not show a single before/after code pair demonstrating how such an “adversarial” edit leads to bad reward signals, unsafe code, or detector evasion in practice; the connection between (L(f(c+\delta),s)) and code quality/safety is not grounded with examples   .
* The detector (f) is unnamed and its training/access assumptions are unclear; transfer to unseen detectors/attacks is not established; key settings ((\epsilon,\lambda), attack steps) and baseline configs are missing, limiting reproducibility and external validity.
* The **presentation** need substaintial improvement. Duplicated subsections and a malformed sentence reduce clarity and trust (e.g., “cannot be used for knifes neckties”; duplicated Section 4.x material).

**Questions:**

See weaknesses

---

### Official Review · Reviewer_tQpj · 2025-11-01

**Soundness:** 2
**Presentation:** 1
**Contribution:** 1
**Rating:** 2
**Confidence:** 3

**Summary:**

This paper presents Adversarial Robust Reward Shaping (ARRS), a reinforcement learning framework designed to generate secure AI code by addressing vulnerabilities to adversarial evasion attacks. It introduces an Adversarial Robustness Module (ARM) that is integrated into the reward pipeline. This module employs gradient-based perturbation analysis to identify worst-case scenarios and applies a robustness penalty to the reinforcement learning policy. The goal of this approach is to optimize both functionality and inherent security, shifting from a reactive to a proactive approach. Experiments conducted on benchmarks such as CodeSearchNet, APPS, and CodeXGLUE were compared against baseline models, showing improvements in robustness metrics, including Attack Success Rate (ASR) and robust accuracy.

**Strengths:**

1. This paper is well structured.
2. Experiments show the effectiveness of ARRS.

**Weaknesses:**

1. The presentation is severely undermined by messy formatting issues, including poorly rendered equations (e.g., repeated and misaligned formulas in Section 4, such as the duplicated robustness penalty derivations), unclear architectural diagrams (e.g., Figure 1's simplistic and unlabeled components fail to clearly illustrate ARM integration), and tables/figures that are large in scope but convey minimal actionable information (e.g., Table 1's sparse ASR comparisons lack statistical significance tests or error bars, while Figures 2 and 3 use generic plots without detailed axes labels or explanations of convergence behaviors).
2. Novelty is limited, as the approach builds heavily on existing adversarial training techniques (e.g., FGSM and PGD from computer vision) adapted to code without substantial innovation.
3. Experimental validation is inadequate: baselines are not fully comparable (e.g., no details on hyperparameter tuning across methods), metrics like vulnerability rates are measured with CodeQL but lack breakdowns by perturbation type.

**Questions:**

Please refer to Weaknesses.

---

### Official Review · Reviewer_K8kQ · 2025-11-01

**Soundness:** 2
**Presentation:** 2
**Contribution:** 2
**Rating:** 2
**Confidence:** 3

**Summary:**

This paper proposes adversarial training for generating secure code that explicitly addresses vulnerabilities to adversarial evasion attacks.

**Strengths:**

Please see below.

**Weaknesses:**

In the background there is a subsection for reinforcement learning for code generation and adversarial robustness for code analysis. One thing came to my mind is that here it might be reasonable to visit research on adversarial and robust reinforcement learning and possibly a small section about it. Because there has been a long line of research that considered gradient based attacks, defenses and detection methods in the reinforcement learning domain [1,2,3,4,5].

Interestingly, there have also been attempts in the reinforcement learning domain to obtain robustness through adversarial training [4]. Another thing is that, more recently some studies demonstrated that adversarial training is vulnerable to black-box adversarial attacks and natural perturbations exhibiting generalization issues in comparison to standard reinforcement learning [2,5].

I am mentioning these concepts from reinforcement learning, because at several points in the paper the exact same concepts have been proposed/re-introduced in a renamed way. For instance, the section titled “Proactive Security by Design in RL Training” the following description is provided:
“Unlike post-hoc filtering, ARM proactively shapes the policy’s behavior by exposing it to adversarial examples during training.” This is called adversarial training.

Given the prior work in reinforcement learning and robustness of it, I am also quite interested if the proposed adversarial training of the submission is robust against more natural perturbations or black-box attacks.

Also regarding experimental results, I have seen more established datasets on coding such as [6,7,8,9,10]. Is there a reason why the experiments did not include these benchmarks? I am only asking this, because prior work [11] seems to test including these mentioned benchmarks.

I think the submission works in a direction that can possibly have promising results. But in its current form it might not be ready to be published yet.

[1] Adversarial Attacks on Neural Network Policies, ICLR 2017.

[2] Adversarial Robust Deep Reinforcement Learning Requires Redefining Robustness, AAAI 2023.

[3] Detecting Adversarial Directions in Deep Reinforcement Learning to Make Robust Decisions, ICML 2023.

[4] Robust Adversarial Reinforcement Learning, ICML 2018.

[5] Deep Reinforcement Learning Policies Learn Shared Adversarial Features Across MDPs, AAAI 2022.

[6] Evaluating Large Language Models Trained on Code, 2021.

[7] Swe-bench: Can language models resolve real-world github issues?, 2023.

[8] Codegeex: A pre-trained model for code generation with multilingual benchmarking on humaneval-x, 2023.

[9] Program synthesis with large language models, 2021.

[10] Competition-level code generation with alphacode, 2021.

[11] CodeT: Code Generation with Generated Tests, ICLR 2023.

**Questions:**

Please see above.

---

### Meta-Review · Area_Chair_mXMH · 2026-01-11

**Summary:**

The paper explores adversarially robust reward shaping for RL-based code generation, integrating an adversarial module into the reward to penalize detector evasion. Reviewers note that the general direction is potentially interesting, and the idea of incorporating worst-case robustness into training is conceptually reasonable, but the strengths are viewed as limited in the current form.

Across reviews, several core concerns remain unresolved. The relationship to prior work in adversarial and robust reinforcement learning is insufficiently clarified, making it difficult to isolate the paper’s substantive contribution beyond reapplication of established adversarial training ideas. Key methodological details (e.g., threat model assumptions, detector specification, hyperparameters, and adversary updates) are underspecified, limiting reproducibility and external validity. Empirical evidence is also considered insufficient, with limited benchmark coverage, unclear baseline comparability, and missing analyses beyond white-box gradient-based attacks. Presentation issues further reduce clarity.

Given the interrupted review process and the lack of explicit post-rebuttal confirmations, this assessment reflects a conservative, best-effort judgment based solely on the available reviews and discussion, without assuming score changes.

**Reviewer Concerns:**

Reviewer K8kQ:

Addressed:
• None

Partially addressed:
• None

Outstanding:
• Positioning vs. prior adversarial/robust RL work remains unclear; core ideas appear overlapping with existing adversarial training literature without sufficient differentiation.
• Robustness beyond white-box, gradient-based attacks (e.g., black-box or natural perturbations) is not demonstrated.
• Limited benchmark coverage without clear justification for excluding established code benchmarks.

Reviewer tQpj:

Addressed:
• None

Partially addressed:
• None

Outstanding:
• Severe presentation and formatting issues (duplicated equations/sections, unclear figures, low-informative tables).
• Limited novelty, largely adapting standard adversarial training methods to code.
• Experimental details insufficient for fair comparison and reproducibility.

Reviewer 7uKR:

Addressed:
• None

Partially addressed:
• None

Outstanding:
• No concrete definition or examples of adversarial code edits demonstrating practical security impact.
• Detector assumptions, threat model parameters, and key hyperparameters are underspecified, limiting reproducibility.
• Presentation issues (duplicated content, malformed text) reduce clarity.

Reviewer 82Rr:

Addressed:
• None

Partially addressed:
• None

Outstanding:
• Scope and boundaries relative to broader LLM attack/jailbreaking literature are insufficiently defined.
• Algorithmic details (e.g., how ARM/adversary is updated during training) are unclear.
• Missing experimental and training details limit reproducibility.

**Reviewer Scores:**

Reviewer K8kQ:

Original score: 2

Likely post-rebuttal score: 2

Justification:
• No explicit reviewer signals indicating satisfaction.
• Major concerns on novelty and robustness generalization remain outstanding.

Reviewer tQpj:

Original score: 2

Likely post-rebuttal score: 2

Justification:
• No post-rebuttal discussion or endorsement.
• Major concerns on presentation, novelty, and experimental rigor remain unresolved.

Reviewer 7uKR:

Original score: 2

Likely post-rebuttal score: 2

Justification:
• No evidence of resolved concerns.
• Core issues on problem grounding, assumptions, and reproducibility persist.

Reviewer 82Rr:

Original score: 2

Likely post-rebuttal score: 2

Justification:
• No explicit positive signals from the reviewer.
• Outstanding concerns on scope, algorithmic clarity, and reproducibility remain.

---

### Decision · Program_Chairs · 2026-01-26

Reject